# Peer review of "The Interplay of Politics and Conspiracy Theories in Shaping Vaccine Hesitancy in a Diverse Cultural Setting in Italy"

_ijerph, 2025, doi:10.3390/ijerph22020230_

Round 1
Reviewer 1 Report
Comments and Suggestions for Authors
Thanks for the invitation to review this manuscript.
In this thought-provoking perspective article, Wiedermann et al. explored the issue of COVID-19 vaccination hesitancy in linked to vaccine conspiracies in Südtirol during the pandemic.
What I found very interesting is the manuscript setting namely Südtirol given its cultural, linguistic, historical distinctive nature compared to the rest of Italy. This unique setting adds importance to this article, since it provides distinctive aspect of COVID-19 vaccine attitudes within a complex sociopolitical setting.
The authors presented a well-written manuscript on how linguistic, cultural, and political aspects intersect with public health behaviors. However, the manuscript primarily draws on literature specific to Italy and Südtirol, which may limit the generalizability of the insights presented. Expanding the scope of literature to involve broader European or global perspectives on COVID-19 vaccine conspiracies can strengthen the manuscript contribution to study topic.
Overall, I endorse this manuscript for publication pending revision of the following points:
Major points
1. In the Introduction, the authors are recommended to provide a more detailed overview of the manuscript context and whether the perspective presented was on COVID-19 vaccination or vaccination in general. Specifically, the authors are recommended to provide an overview of vaccination attitudes in Südtirol in terms of parental attitude to childhood vaccination, influenza vaccination attitude among health professionals, etc.
2. The Discussion appeared to reiterate what has already been presented in the previous sections. I recommend re-organizing this section to focus more on policy implications and strategic recommendations that can offer actionable insights for improving vaccination attitudes in Südtirol. Emphasizing practical strategies, such as culturally tailored health communication or targeted community engagement, could make the conclusions more impactful. Additionally, to enhance the broader applicability of the perspective findings, the authors are recommended to consider framing their insights within a wider regional context. Drawing comparisons with other culturally or linguistically diverse regions in Europe could provide a more comprehensive understanding of how similar factors influence COVID-19 vaccine hesitancy across borders.
3. It is unclear how the authors conducted their literature review. I guess it was based on their previous study: https://doi.org/10.7416/ai.2024.2625. If so, the authors are recommended to briefly state their literature search approach and how this could be a limitation in this perspective.
4. A possible limitation of the manuscript is that the authors may have approached covid-19 vaccine conspiracies with a negative connotation, which could introduce bias in the framing and interpretation of the findings. While harmful misinformation exists, it is important to avoid oversimplifying vaccine hesitancy as merely a result of exploitation by certain political groups. The general public concerns often stem from poor risk communication, inconsistent public health messaging, and historical mistrust, all of which warrant deeper examination in the context of Südtirol. Addressing these factors with a more balanced discussion could strengthen the manuscript objectivity and offer a more comprehensive understanding of COVID-19 vaccine attitudes in Südtirol.
Minor points
1. Please follow the IJERPH guidelines regarding the structure of the Abstract as listed in: https://www.mdpi.com/journal/ijerph/instructions “The abstract should be a total of about 200 words maximum. The abstract should be a single paragraph and should follow the style of structured abstracts, but without headings…”
2. In the Introduction (lines 32 – 33): For the definition of vaccine hesitancy, please cite the seminal work of Noni E MacDonald in: https://doi.org/10.1016/j.vaccine.2015.04.036 and the notable work of Patrick Peretti-Watel, Heidi J Larson et al. in: https://pmc.ncbi.nlm.nih.gov/articles/PMC4353679/
3. Line 44: please mention that the study was related to COVID-19 vaccine hesitancy rather than vaccine hesitancy in general. This is important given the specific nature of vaccination hesitancy.
4. Lines 80 – 81: I think that it is better to revise the sentence “These linguistic groups have historically shaped health behaviours, including vaccine acceptance” as follows: Linguistic groups have historically shaped health behaviors, including attitudes toward vaccine acceptance. This revision is needed since you cited a reference from Austria and you can add further references to support this premise.
5. Lines 155 – 158: Please add COVID-19 since the major talking points that led to this summary was in the context of COVID-19 vaccination.
6. For Figure S1 in the supplementary file, please check if there are any copyright issues.
Best wishes!
Reviewer 2 Report
Comments and Suggestions for Authors
Dear Author’s.
Historically, in South Tyrol, resistance to vaccinations has been long-standing. The population is not uninformed, but informed and ready to support its position. It is a population that believes in phytotherapy, practices it and passes it on to its generations. The German-speaking population is strongly resistant to vaccinations, they live in groups and do not like to publicize their beliefs. Even pediatric vaccinations are not practiced and children live in a different reality and are often educated outside of public institutions. German-speaking children often live outdoors and eat naturally and parents do not feel the need to vaccinate because they are convinced that vaccines can cause adverse events, even serious ones.
Furthermore, parents rarely resort to common pharmacological treatments and try to preserve the intestinal microbiota. Therefore, it is a population that cannot be influenced by the mass media and health authorities, especially if they belong to the central government.
Those who publish this type of article commit the fundamental error of considering these people as uninformed, since they do not uncritically accept the health message coming from the promoters of vaccinations. Here there is no possibility of producing “positive frames” within which to enclose the vaccination message. These populations do not voluntarily access alternative information with respect to their historical and social experience.
Monitoring information is perceived as censorship and the behavior of the Italian health authorities in the recent SARS-CoV-2 pandemic has widened the distance between health authorities and the population. All opinion leaders exposed in the mass media during the recent pandemic have contributed to widening the distance. The sources of scientific information to which these populations resort are not Italian, they come from the German world with which they maintain fundamental relations.
However, the fundamental element that prevents a dialogue between the population and central health authorities is the high level of distrust in public institutions and scientific authorities. Furthermore, I think that the “irritating” element of “disinformation” should be eliminated from this study because it is not true that they are misinformed, they access alternative sources of information, they access adverse event registration systems and compare these data with the phrase “the vaccine is effective and safe”, revealing a dissonance between tangible data (adverse events always possible and registered in VAERS) and unrealistic guarantees (absence of adverse events because vaccines are effective and safe).
If health authorities want to become a reliable source of information, they must be transparent, they must communicate that post-vaccination adverse events are possible, and they must build a real risk/benefit balance of vaccinations. Furthermore, they must not rely on communicators who claim to be experts without a specific curriculum in vaccinology. Finally, making the interlocutor understand that he is a conspiracy theorist and/or uninformed is harmful, since these are often informed and skeptical people who live in symbiosis with nature in minimally contaminated environments.
In short, there is no possible action to remove the refusal to vaccinate because it is a cultural attitude that has developed in a population that gets its information outside of Italy, does not trust the central health authorities and does not use social media to build its own beliefs about vaccinations.
Best regards.
Author Response
Thank you for your thoughtful and detailed feedback. In response to your comments, we have made several revisions to the manuscript:
-
Historical Resistance to Vaccination – We have expanded the Introduction to include a discussion of South Tyrol’s long-standing skepticism toward vaccinations, emphasizing the region’s reliance on alternative medicine and natural health practices.
-
Alternative Knowledge Systems – In Section 4.1, we have clarified how vaccine hesitancy in South Tyrol is shaped not merely by disinformation but by trust in alternative knowledge sources, particularly non-Italian health information. Additionally, we have acknowledged the role of cultural factors, such as the preference for natural immunity and traditional health practices, in reinforcing hesitancy.
-
Credible Communicators – We have added a discussion on the importance of engaging trusted local figures, such as community doctors, midwives, and alternative health practitioners, in vaccine communication strategies to improve public confidence.
These changes aim to better contextualize vaccine skepticism in South Tyrol and refine our discussion on public health messaging strategies. We appreciate your valuable insights.